# Single-nucleus transcriptomic analysis of human dorsal root ganglion neurons

**Minh Q Nguyen[1], Lars J von Buchholtz[1], Ashlie N Reker[2], Nicholas JP Ryba[1]\*, Steve Davidson[2]\***

[1]National Institute of Dental and Craniofacial Research, National Institutes of Health, Bethesda, United States; [2]Department of Anesthesiology, College of Medicine, University of Cincinnati, Cincinnati, United States

**Abstract** Somatosensory neurons with cell bodies in the dorsal root ganglia (DRG) project to the skin, muscles, bones, and viscera to detect touch and temperature as well as to mediate proprioception and many types of interoception. In addition, the somatosensory system conveys the clinically relevant noxious sensations of pain and itch. Here, we used single nuclear transcriptomics to characterize transcriptomic classes of human DRG neurons that detect these diverse types of stimuli. Notably, multiple types of human DRG neurons have transcriptomic features that resemble their mouse counterparts although expression of genes considered important for sensory function often differed between species. More unexpectedly, we identified several transcriptomic classes with no clear equivalent in the other species. This dataset should serve as a valuable resource for the community, for example as means of focusing translational efforts on molecules with conserved expression across species.

## Editor's evaluation

The manuscript by Nguyen et al. describes the assignment of neuronal cell types to human lumbar dorsal root ganglion (DRG) neurons based on the sequencing of individual nuclei. Bioinformatic comparison of these data to single nucleus sequencing results previously reported for mouse lumbar DRG is also described. The findings begin to close a gap in our understanding of how neuronal cell types and the expression of key genes differs between human and mouse. This kind of information is critical for targeted therapeutic efforts.

**\*For correspondence:**
nick.ryba@nih.gov (NJPR);
davidsst@ucmail.uc.edu (SD)

**Competing interest:** The authors declare that no competing interests exist.

## Introduction

The somatosensory system responds to a wide range of mechanical, thermal, and chemical stimuli to provide animals with critical information about their environment and internal state. For example, our sense of touch is mediated by mechanosensory neurons with soma located in the dorsal root and trigeminal ganglia that innervate the skin (*Abraira and Ginty, 2013*). In addition to the skin, somatosensory neurons target specialized sensory environments like the cornea and conjunctiva or meninges (*von Buchholtz et al., 2020*; *Huang et al., 2018*), the internal organs (*Hockley et al., 2019*) as well as bones and muscles to provide rich perceptual experiences and trigger appropriate behavioral, reflex, and autonomic responses (*Gatto et al., 2019*). Among their many roles, somatosensory neurons provide input for the conscious perception of pain and itch (*Basbaum et al., 2009*; *Mishra and Hoon, 2013*) and the subconscious coordination of muscles and limbs known as proprioception (*Chesler et al., 2016*). Peripheral neurites of somatosensory receptor cells must adapt to growth, reinnervate targets after injury and are also affected by inflammation (*Pongratz and Straub, 2013*).

Studies in model organisms have characterized a range of sensory and growth factor receptors and ion channels that contribute to the properties and selectivity of somatosensory neurons (*Le Pichon and Chesler, 2014*; *Coste et al., 2010*; *Ranade et al., 2014*). Some of these, like the cooling and menthol sensing receptor (*Trpm8*) appear to define functional classes of cells (*Bautista et al., 2007*). By contrast, the sense of touch appears to use a complex distributed code involving several different types of cells (*von Buchholtz et al., 2021b*) to achieve its remarkable discriminatory power. For the most part, the human somatosensory system expresses the same range of functional genes as rodents (*Ray et al., 2018*) and exhibits similar responses to many types of stimulus (*Adriaensen et al., 1983*; *Basbaum et al., 2009*; *Chesler et al., 2016*; *Le Pichon and Chesler, 2014*; *Ranade et al., 2014*; *Bautista et al., 2014*; *Davidson et al., 2016*). Moreover, rare individuals with loss of function variants of several of these genes have deficits that recapitulate key effects of knocking out that gene in mice (*Chesler et al., 2016*; *Murthy et al., 2018*; *Szczot et al., 2018*; *Drenth and Waxman, 2007*; *Chen et al., 2015*). However, despite the identified similarities between mice and humans, the success of translating new therapeutic strategies that are effective for treating pain in mice has often been disappointing when tested in human subjects (*Mogil, 2019*; *Yezierski and Hansson, 2018*).

Recently, various directed genetic strategies have been used in mice to characterize the response properties and anatomical features of a variety of interesting classes of large diameter, fast conducting Aβ and Aδ subtypes (*Abraira and Ginty, 2013*). Interestingly, these neurons generally have complex peripheral endings that often target hair follicles. Human skin hairs are quite different from those in mice, suggesting that there may be significant differences between the large diameter neurons in mice and humans. By contrast, most types of small diameter, slow conducting c-fibers terminate as free nerve endings both in mice and humans (*Basbaum et al., 2009*). Single-cell sequencing approaches have produced a transcriptomic classification for mouse somatosensory neurons that corresponds well with their anatomy and function (*Gatto et al., 2019*; *von Buchholtz et al., 2021b*; *Sharma et al., 2020*; *Nguyen et al., 2019*). In mice, at least two classes of small diameter neurons are best defined by different members of the Mrgpr family of GPCRs (*Sharma et al., 2020*; *Nguyen et al., 2019*). However, Mrgprs have undergone massive genetic expansion in rodents, not seen in other animals, often making it difficult to identify true orthologs in humans (*Dong et al., 2001*; *Liu et al., 2009*) and raising questions as to whether similar cells would have distinct molecular markers in the two species. A map of human somatosensory neuron transcriptomic classes would help uncover selective differences between the sensory neurons in mice and humans and provide clues as to how similar somatosensory input is in the two species. Finally, such analysis may provide important new targets to consider for translational approaches to treat both pain and itch. Here, we used nuclei-based single-cell transcriptomics to generate an initial description of human cell types, highlight similarities and surprising differences between somatosensory neuron classes in humans and mice that are reflected not only in terms of individual genes but can be discerned in co-clustering. We used multigene in situ hybridization (ISH) to help confirm these conclusions and present evidence for anatomic organization of functionally distinct neuronal classes in the human dorsal root ganglion.

## Results

### Generating a representative transcriptomic map of human somatosensory cell types

Single lumbar L4 and L5 human dorsal root ganglia (DRG) were rapidly recovered from transplant donors within 90 min of cross-clamp and were immediately stored in RNAlater. Nuclei from individual ganglia were isolated and samples were enriched for neuronal nuclei by selection using an antibody to NeuN (see *Figure 1—figure supplement 1*). Five ganglia from one male and four female donors with ages ranging from 34 to 55 were subjected to droplet based single nucleus (sn) capture, barcoding, and reverse transcription (×10 Genomics). Combinatorial clustering methods (*Stuart et al., 2019*) allowed co-clustering of neuronal nuclei into a well-defined set of distinct transcriptomic groups that are well separated from their non-neuronal counterparts (*Figure 1A*, *Figure 1—figure supplement 2*). After removal of non-neuronal nuclei from the dataset, reclustering the DRG-neuron data from 1837 cells identified a range of about a dozen diverse transcriptomic classes of human somatosensory neurons (*Figure 1B*, *Figure 1—figure supplements 2 and 3*).

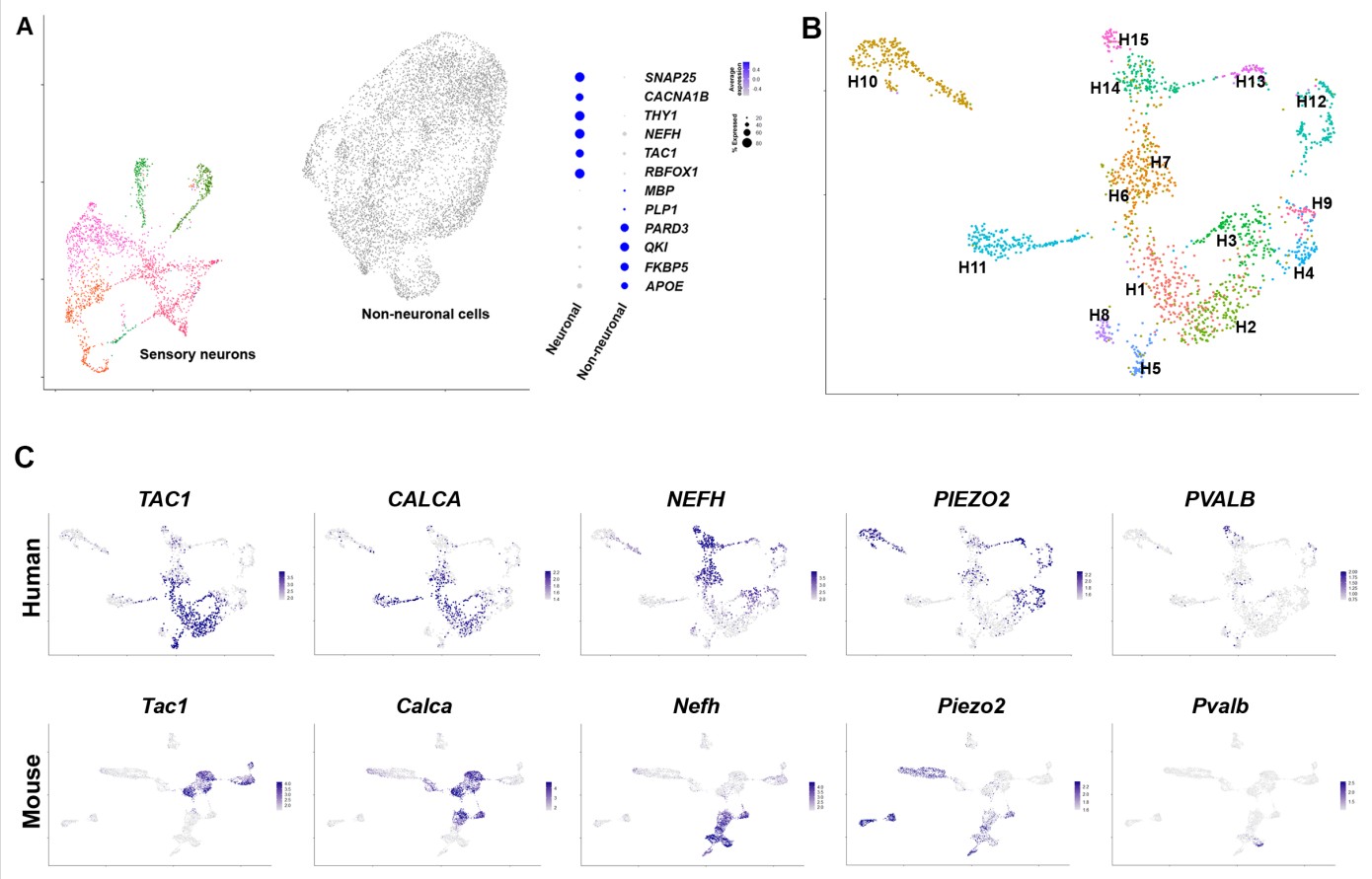

**Figure 1.** Diverse classes of human dorsal root ganglia (DRG) neurons revealed by single nuclear transcriptomics. (**A**) Universal manifold (UMAP) representation of graph-based co-clustered snRNA sequences from human DRG nuclei reveal two well separated groups corresponding to sensory neurons (colored) and non-neuronal cells (gray). To the right, a dotplot highlights the expression of markers that help distinguish these groups of cells (see also *Figure 1—figure supplement 2A* for more information about preliminary analysis). (**B**) Reanalyzing 1837 neuronal nuclei clusters human DRG neurons into transcriptomically distinct groups that have been differentially colored. (**C**) Similarity in expression of differentially expressed genes between human and mouse neuronal types may help functional classification of neuronal types: UMAP representation of human DRG neurons showing relative expression level (blue) of diagnostic markers. For comparison UMAP representation of mouse neurons (*Renthal et al., 2020*) showing the relative expression patterns of the same markers. In combination, the expression patterns of these and other genes (*Figure 1—figure supplements 2 and 4*, *Figure 1—figure supplement 5*) were used to tentatively match several human and mouse transcriptomic classes (*Figure 1—figure supplement 4*).

The online version of this article includes the following figure supplement(s) for figure 1:

**Figure supplement 1.** Human dorsal root ganglia (DRG) neurons express NeuN.

**Figure supplement 2.** Support for the clustering of human dorsal root ganglia (DRG) neurons.

**Figure supplement 3.** Robust clustering of human dorsal root ganglia (DRG) neurons.

**Figure supplement 4.** Additional markers that support similarities between dorsal root ganglia (DRG) neuronal clusters across species.

**Figure supplement 5.** Dotplots of gene expression supporting similarity of several dorsal root ganglia (DRG) neuronal classes in mice and humans.

One of the best studied groups of somatosensory receptors in mice are nociceptive peptidergic neurons that coexpress a variety of neuropeptides including substance P, calcitonin gene-related peptide (CGRP), and pituitary adenylate-cyclase-activating polypeptide (PACAP). These neurons are typically small soma diameter, nonmyelinated, slow conducting c-fibers, but also include faster conducting lightly myelinated Aδ neurons (*Sharma et al., 2020*; *Nguyen et al., 2019*). In the human DRG dataset, *TAC1* (substance P), *CALCA* and *CALCB* (CGRP), and *ADCYAP1* (PACAP), are expressed in several transcriptomic classes (H1, H2, H3, H5, and H6, *Figure 1C*, *Figure 1—figure supplements 4 and 5*). For comparison the expression of the same genes in mouse DRG neurons is shown (*Figure 1C*, *Figure 1—figure supplements 4 and 5*) using data from single nuclei sequencing (*Renthal et al.,*

*2020*). Just as in mice, the putative human peptidergic nociceptors express the high affinity nerve growth factor receptor *NTRK1*, the capsaicin and mustard oil-gated ion channels *TRPV1* and *TRPA1* but generally only low levels of the stretch-gated ion channel *PIEZO2* (*Figure 1C*, *Figure 1—figure supplements 4 and 5*).

Although previous localization studies have suggested that in humans the neurofilament protein *NEFH* is expressed in all sensory neurons (*Rostock et al., 2018*), this gene showed graded expression in our data (*Figure 1C*) and marks several classes of cells just as in mice (*Figure 1—figure supplement 5*). Some of these (including H3 and H6) also express peptidergic markers and the pain-related voltage-gated sodium channel *SCN10A* (*Figure 1—figure supplement 5*) and thus have molecular hallmarks of Aδ nociceptors (*von Buchholtz et al., 2020*). However, the neuronal classes H14 and H15 expressing the highest levels of *NEFH* are distinct from the peptidergic neurons (*Figure 1C*, *Figure 1—figure supplements 4 and 5*), likely representing different types of large diameter, fast conducting myelinated Aβ neurons. These cell types are neurotrophin three receptor *NTRK3* positive, some also contain the brain derived neurotrophic factor receptor *NTRK2* but exhibit little expression of *NTRK1* (*Figure 1—figure supplements 4 and 5*). In mice, proprioceptors are a subtype of Aβ neurons marked by the calcium binding protein parvalbumin, the transcription factor *Etv1* and the voltage-gated sodium channel subunits *Scn1a* and *Scn1b* (*Sharma et al., 2020*; *Renthal et al., 2020*). In the human data, the small H15 group of *NTRK3*-positive cells had this expression pattern (*Figure 1C*, *Figure 1—figure supplement 5*) implying that proprioceptors have conserved transcriptomic markers in humans and mice. Similarly, small groups of both Aδ-low threshold mechanosensors (H13) and cool responsive neurons (H8) were identified by their characteristic expression profiles of functionally important transcripts (*Figure 1—figure supplement 5*). Thus, large groups of human and mouse DRG neurons appear to share basic transcriptomic signatures and functional potential, supporting our data as informative about neuronal diversity among human somatosensory neurons.

Despite these broad similarities between the putative peptidergic, proprioceptive, cooling sensitive, Aβ and Aδ classes of DRG neurons in mice and humans there were differences in expression of many genes. These include molecules that modulate cellular responses to internal signals (e.g., growth factor receptors), sensory stimuli and also the mediators they may release. For example, in humans, the H8 putative cool responsive neurons expressing *TRPM8* were strongly positive for the BDNF-receptor *NTRK2* but hardly expressed the neuropeptide *TAC1* whereas in rodents the converse was true (*Figure 1—figure supplement 5*). Other genes that have been shown to control sensory responses in mice exhibit a different expression pattern in human DRG neurons. For instance, *Tmem100* encodes a protein that in mice has been implicated as playing an important role in functional interactions between *Trpv1* and *Trpa1* and contributing to persistent pain (*Weng et al., 2015*). By contrast it was almost undetectable in the human sequencing data (*Figure 2A*). Similarly, we did not detect marked expression of the sphingosine-1-phosphate receptor *S1PR3* (*Figure 2A*) that has been suggested as a target for treating both pain and itch based on mouse work (*Hill et al., 2018*). More strikingly, a small group of human neurons, H5, expressing *TRPA1* were resolved in our clustering (*Figure 2B*), whereas in mouse nuclear sequencing data no direct counterpart was detected (*Figure 1—figure supplement 4*). Whole cell-based sequencing (*Sharma et al., 2020*) of mouse DRG neurons does identify a group of peptidergic nociceptors (called CGRP-gamma) with abundant *Trpa1* expression, highlighting the caution needed when interpreting differences across species and the value of alternative approaches. Nonetheless, whereas mouse CGRP-gamma neurons strongly express *Calca* and *Ntrk1*, H5 cells are essentially *CALCA* (CGRP) and *NTRK1* negative and instead are strongly *NTRK2* positive (*Figure 1*, *Figure 1—figure supplements 4 and 5*) suggesting that they may respond differently to external stimuli as well as in their signaling properties and therefore may not be a direct equivalent of mouse CGRP-gamma neurons. Thus, the availability of human transcriptomic data should help focus translational work in model organisms on promising targets with conserved expression patterns in humans.

## Human DRG neurons without clear transcriptomic equivalents in mice

Analysis of the gene expression patterns of the different classes of human somatosensory neurons revealed several groups for which we could not discern direct counterparts in the mouse. One small but prominent group of human DRG neurons (H9) expresses *TRPM8*, *PIEZO2*, *SCN10A*, and *SCN11A* (*Figure 1—figure supplements 4 and 5*, *Figure 2B*) and clearly segregates from the putative cool

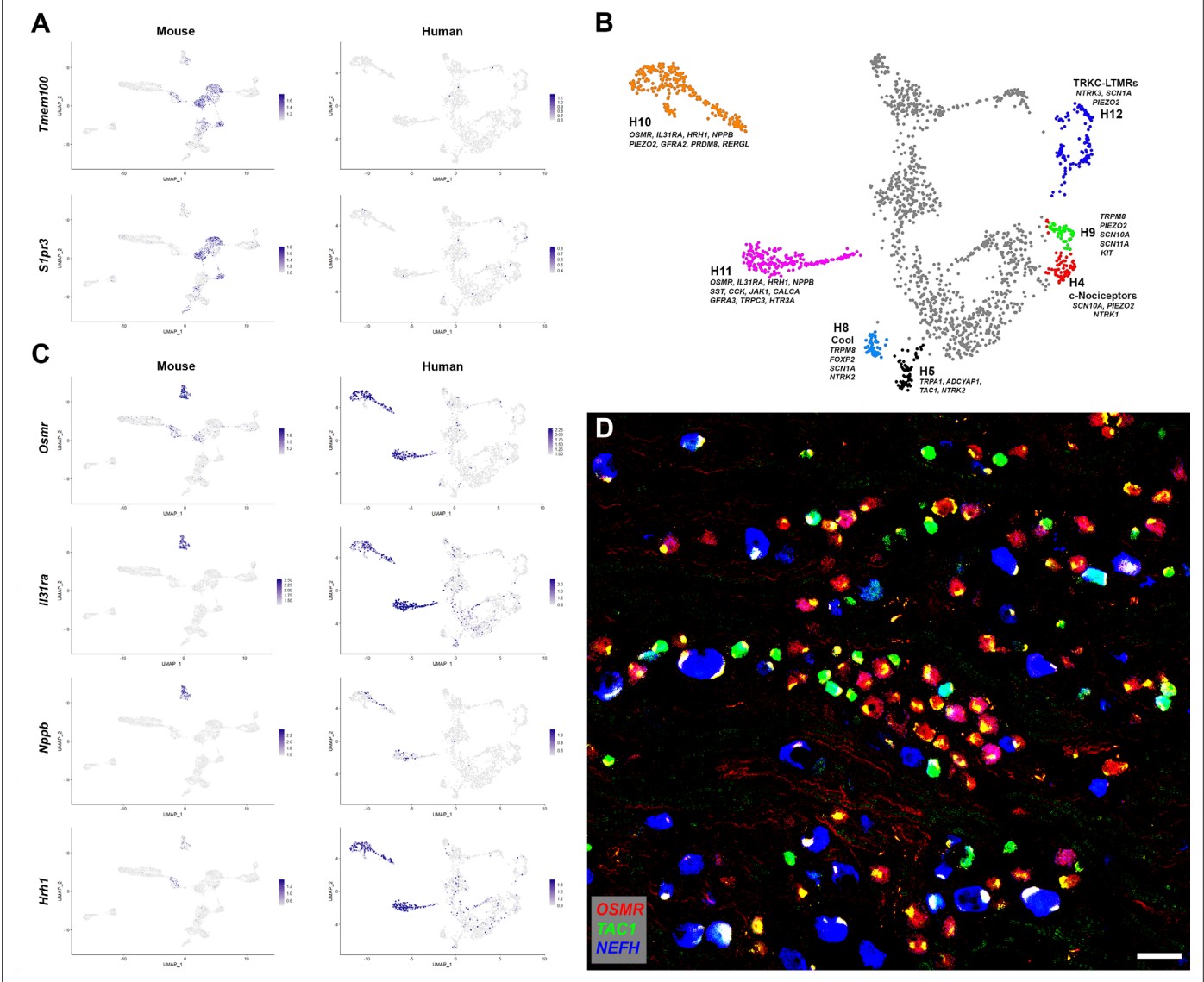

**Figure 2.** Human DRG neurons exhibit specialization that distinguishes them from mouse counterparts. (**A**) Universal manifold (UMAP) representation of mouse and human dorsal root ganglia (DRG) neurons showing relative expression level (blue) of two genes that have been linked to pain sensation in mice. Note that both *TMEM100* and *S1PR3* are more sporadically expressed by the human somatosensory neurons. (**B**) Classes of DRG neurons that are selectively detected in humans are highlighted together with their expression of key genes. H9 neurons coexpress the cool and mechanosensory ion channels; for comparison cool sensitive neurons (H8) that correspond more closely with their rodent counterparts are also highlighted. (**C**) Expression profiles of several itch-related genes in the mouse and human DRG transcriptome. (**D**) Confocal image of a region from a human DRG that was labeled using multiplexed in situ hybridization (ISH) for *OSMR*, *TAC1*, and *NEFH* as indicated in the key. Almost all neurons were *OSMR*, *TAC1*, or *NEFH* positive (***Figure 2—figure supplement 1***). However, few neurons were strongly positive for more than one of these markers (see ***Figure 2—figure supplement 2*** for individual channels). Note that autofluorescence in all channels from lipofuscin associated with many human neurons should not be confused with real signal (see ***Figure 2—figure supplement 2*** for more detail). Also note that *NEFH* is typically expressed in larger diameter neurons than the other two markers. Scale bar = 100 μm.

The online version of this article includes the following figure supplement(s) for figure 2:

**Figure supplement 1.** Quantification of in situ hybridization (ISH) data.

**Figure supplement 2.** H10 and H11 are classes of human dorsal root ganglia (DRG) neurons that express a range of itch-related genes.

sensing cells (H8) that express *TRPM8*, *GPR26*, *NTM*, and *FOXP2* but are devoid of both the light touch receptor and the pain-related sodium channels (***Figure 2B***, ***Figure 1—figure supplement 5***). In mice, *Trpm8* expression is simpler with the cool sensing, menthol responsive ion channel just expressed in cells with this latter gene expression pattern (***Figure 1—figure supplement 5***). Interestingly, single

fiber recordings have identified human neurons that respond to both cooling and gentle touch as might be expected for cells expressing both *TRPM8* and *PIEZO2* (*33*). Moreover, recent single-cell sequencing of macaque DRG neurons also identified two populations of cells that match H8 and H9 neurons (*Kupari et al., 2021*). Finally, H9 neurons resemble (but also have differences from) human mechanosensory neurons that were recently engineered by transcriptional programming of stem cells (*Nickolls et al., 2020*).

A second larger group of human neurons H12 is marked by *NTRK3* and the voltage-gated ion channel *SCN1A*, but is only weakly positive for *NEFH*, expresses moderate levels of *PIEZO2* (*Figure 2B*, *Figure 1—figure supplement 5*) and appears distinct from any potential mouse counterpart. The H12 gene expression pattern is most consistent with these cells functioning as a type of mechanosensor that has no direct equivalent in mice. Similarly, we designated H4 as c-nociceptors because of their expression of nociception-related *SCN10A* and *NTRK1* and low level of *NEFH* (*Figure 2B*, *Figure 1—figure supplement 5*). These neurons expressed low levels of neuropeptides, but their overall gene expression patterns did not resemble any mouse counterparts including the nonpeptidergic nociceptors (see below).

The two remaining large groups of neurons in the human dataset H10 and H11 that have no clear mouse counterpart exhibit most similarity with mouse c-type nonpeptidergic neurons (*Figure 2—figure supplement 2A*). At a functional level both H10 and H11 express receptors that in mice have roles in detecting pruritogens. For example, these clusters were positive for the two subunits (*IL31RA* and *OSMR*) of the interleukin 31 receptor and the histamine receptor *HRH1* (*Figure 2C*) that mediate mast cell-related scratching in mice (*Solinski et al., 2019*). They also express the itch-related neuropeptide *NPPB* (*Figure 2C*), nociception-related sodium channels *SCN10A* and *SCN11A* as well as *TRPV1* (*Figure 2—figure supplement 2A*) but not appreciable *NEFH* or *TAC1* (*Figure 1C*). Therefore, it is likely that these are groups of putative unmyelinated, nonpeptidergic nociceptors with potential for triggering human itch responses. H10 and H11 characterization is revisited later in its own section of the results.

The peptidergic nociceptors, myelinated Aβ and Aδ neurons, rarer human-specific cells, and the two nonpeptidergic nociceptor clusters H10 and H11 account for all the neurons in our analysis with H10 and H11 totaling approx. 20% of the neurons. In marked contrast, mouse nonpeptidergic, small diameter neurons are far more numerous than H10 and H11 accounting for 40% of the sensory neurons in mouse DRGs (*Renthal et al., 2020*) and divide into four highly stereotyped transcriptional groups (*Figure 1—figure supplement 4*). Two of these classes of mouse neurons (NP2 and NP3) trigger itch (*Mishra and Hoon, 2013*; *Han et al., 2013*), one (NP1, expressing *Mrgprd*) responds to noxious mechanical stimulation (*von Buchholtz et al., 2021b*). NP1 neurons may have a role in mechanonociception (*Gatto et al., 2019*) and have recently been associated with suppression of skin inflammation (*Zhang et al., 2021*), which was hypothesized as relevant for human health. The fourth class corresponds with low threshold mechanosensors (cLTMRs) that are thought to mediate affective touch (*Gatto et al., 2019*; *McGlone et al., 2014*). Given this difference between the transcriptomic map of human DRG neurons and their rodent counterparts, we next used independent ISH-based analysis to test basic predictions of the sequencing. If transcriptomic characterization of human DRG neurons is accurate then one clear expectation is that *TAC1*, *NEFH*, and *OSMR* should be expressed by distinct and only partially overlapping populations of human DRG neurons. If it is also comprehensive, that is, not missing equivalent classes to mouse neurons NP1, NP2, and cLTMRs that constitute almost a third of mouse DRG neurons and do not significantly express *Nefh*, *Tac1*, or *Osmr*, then we would anticipate that the same three markers should label the vast majority of neurons. Multigene ISH demonstrates that both these predictions are true for human DRG neurons (*Figure 2D*, *Figure 2—figure supplements 1B and 2B*) with essentially every cell labeled by one of these probes but with very few exhibiting strong coexpression. Although *NEFH* expression could be detected in some of the cells positive for the other markers (*Figure 2D*, *Figure 2—figure supplement 1B*), many *TAC1*- or *OSMR*-positive small diameter neurons were negative for this neurofilament subunit. Moreover, *TAC1* and *OSMR* labeled almost completely separate sets of cells (*Figure 2—figure supplement 1*). Notably, in keeping with our assignments based on transcriptomic data, the largest diameter neurons were strongly positive for *NEFH* whereas *TAC1* and *OSMR* primarily labeled smaller cells (*Figure 2D*). Finally, these three markers each labeled a large group of neurons (*Figure 2—figure supplement 1*).

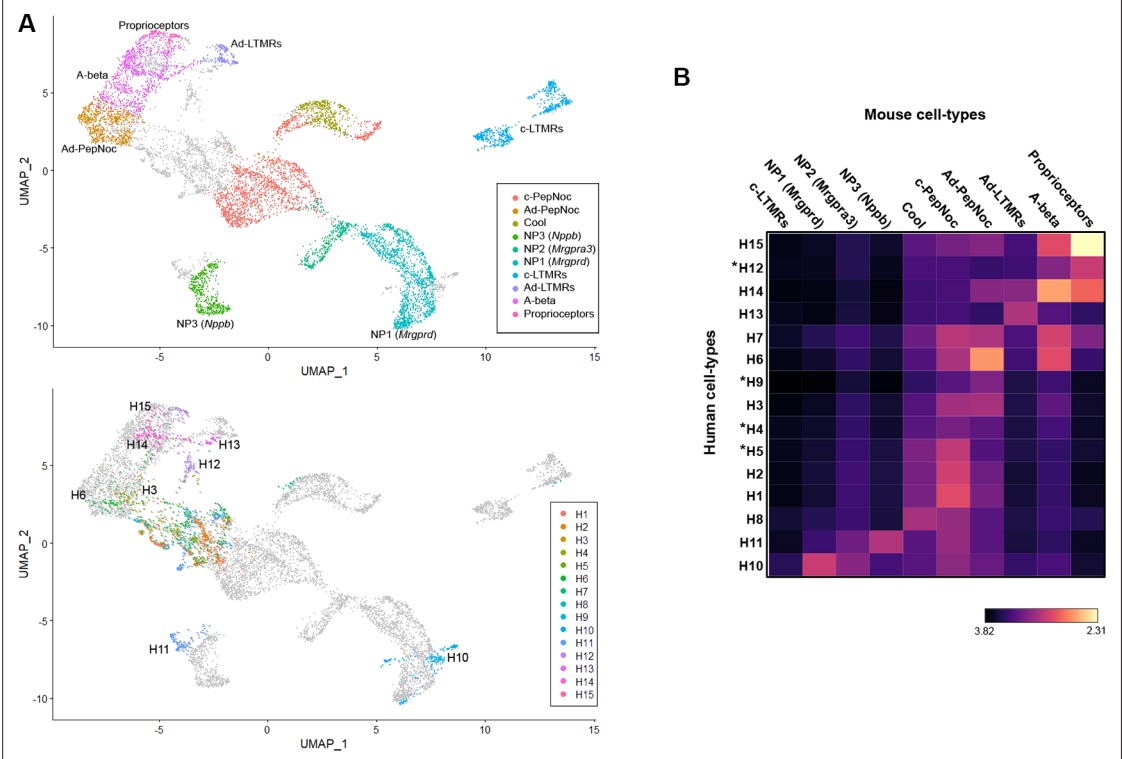

**Figure 3.** Co-clustering of human and mouse neurons support tentative assignments based on select genes. (**A**) Universal manifold (UMAP) representation of the co-clustering of mouse and human neurons. Upper panel shows the mouse neurons colored by their identity when analyzed alone (*Figure 1—figure supplement 1*); lower panel shows human neurons colored by their identity when analyzed alone (*Figure 1*). Note that large diameter human neurons match their expected mouse counterparts reasonably well and that the two classes of neurons expressing itch-related transcripts H10 and H11 best match NP1 and NP3 neurons, respectively. (**B**) Heatmap showing the natural logarithm (see scale bar) of Kullback–Leibler divergences for the various human neuron classes when compared to each class of mouse cells as a reference distribution; potentially human-specific classes based on functional markers are marked by *; see also *Figure 3—figure supplement 1*.

The online version of this article includes the following figure supplement(s) for figure 3:

**Figure supplement 1.** More detailed analysis of relationships between human and mouse cell types.

## Co-clustering human and mouse DRG-neuron snRNAseq data

As detailed above, the expression of genes that are thought to be important for functional and morphological features of somatosensory neurons reveal similarities between groups of human and mouse neurons. They also expose differences that likely reflect distinct somatosensory adaptations in the two species. We next used co-clustering methods to test whether the wider transcriptome could reveal additional information about the relationships between classes of human and rodent DRG neurons using the same mouse dataset (*Renthal et al., 2020*) that we analyzed above (*Figure 1*, *Figure 1—figure supplement 4*). We used the well-established approach developed by the Satija lab (*Stuart et al., 2019*) as it has been shown to perform well without forcing false matches. As predicted, several classes of human neurons grouped with corresponding mouse counterparts including H15 – proprioceptors, H14 – Aβ cells, H13 – AδLTMRs, H11 – NP3 (*Nppb*) neurons, and H3/H6 – Aδ nociceptors (*Figure 3A*). This analysis suggested that H10 the other cluster that gene expression indicated are also itch related most closely resembled NP1 (*Mrgprd*) neurons rather than any other human or mouse class of sensory neurons. The H12 cluster, which is human specific, grouped close to larger diameter mouse neurons, whereas other clusters of human cells appeared better aligned with smaller diameter nociceptors. However, all types of peptidergic small diameter nociceptors were less organized in the co-clustering and separated from their potential mouse counterparts despite their qualitatively similar expression of some of the best-known functional markers (*Figure 1—figure supplement 4*).

UMAP plots (*Figure 3A*) provide a visual representation of similarity between cells with related transcriptomic properties. However, co-clustering methods do not provide a quantitative measure

of the similarity between individual cells or clusters of cells. A given cell cluster can be viewed as a multivariate probability distribution in gene expression space. While not commonly employed in gene expression analysis, in probability and information theory, the similarity/dissimilarity between two probability distributions is most commonly measured by calculating their Kullback–Leibler (KL) divergence. Recent advances have allowed KL divergence to be estimated between two samples in continuous multivariate space (*Perez-Cruz, 2008*) as is observed in dimensionality reduced gene expression data of cell populations. Therefore, we made use of KL divergence estimation to quantitate the similarity between human DRG-neuron clusters and all their potential mouse counterparts (*Figure 3B*). As expected, clusters that co-segregate in the UMAP analysis showed greatest similarity but additional relationships not apparent from the visual representation of the co-clustering were also seen. For example, the small cluster of human 'cool' responsive neurons H8 showed greatest similarity to mouse Trpm8 cells and several groups of human cells (H1, H2, and H5) that gene expression predicted should be c-type peptidergic nociceptors, indeed best matched these cells (*Figure 3B*). Interestingly, no class of human neurons showed appreciable similarity to mouse cLTMRs. Among the groups of cells that had human-specific gene expression patterns, H5 best matched c-peptidergic nociceptors, H9 (the putative cool and mechanical responsive cells) showed only weak similarity to any mouse neuron class. H12, which we considered likely to be mechanosensors best matched mouse proprioceptors and H4 neurons appeared distantly related to several classes of nociceptor but without a clear match in mice. We also extended this analysis to mouse clustering where similar cell populations had not been combined (*Figure 3—figure supplement 1*). Although 19 clusters of mouse neurons were now analyzed, neither mouse cLTMRs nor the potentially human-specific classes H4, H9, or H12 better matched a cell types of the other species. One important caveat to this type of analysis remains that any functional conclusions based on shared transcriptomic features still need to be verified experimentally.

## Transcriptomically related neurons are spatially grouped in the human dorsal root ganglion

From sequence analysis, we identified a range of potential markers to better explore the diversity of human DRG neurons using ISH. To maximize information, we chose a multiplexed approach that allows localization of up to 12 probes (*Figure 4*) revealing the different classes of sensory neurons identified in the transcriptomic data. For example, *TRPM8* expressing neurons clearly segregate into two distinct types (*Figure 4A*, *Figure 4—figure supplement 1*). One set of cells (H8) share other transcriptomic properties with mouse cooling responsive cells: *TRPM8*, the cool and menthol receptor is not coexpressed with the ion channels *SCN10A* or *PIEZO2* (*Figure 4A*), but unlike in mice these cells are *NTRK2* positive (*Figure 4—figure supplement 1*). By contrast, other cells (H9) coexpress the pain and light touch related ion channels (*SCN10A* and *PIEZO2*) with *TRPM8* (*Figure 4A*, *Figure 4—figure supplement 1*). Similarly, putative proprioceptive neurons (H15) were distinguished by their expression of *NEFH*, *PIEZO2*, and *PVALB* and lack of *NTRK2* (*Figure 4B*, *Figure 4—figure supplement 1*). One surprise (*Figure 4A, B*) was that in small fields of view, several examples of all three of these rare neuron types could be identified in human DRGs. However, much of the rest of the ganglion was devoid of these cell types and instead the neurons there had distinct sets of markers. Therefore, it appears that transcriptomic classes of human DRG sensory neurons may not be stochastically distributed in the ganglion. Indeed, when we examined the distribution of nociceptors and myelinated neurons at lower magnification (using strong selective probes), broad clustering of similar types of neurons was apparent, quantifiable, and statistically significant (*Figure 4C, D*). We carried out a similar analysis in mouse DRG neurons using *Nefh* and *Scn10a* probes and found that there too, cells expressing these markers are not uniformly distributed (*Figure 4—figure supplement 2*) suggesting spatial clustering of related DRG neurons may be a common feature across species.

## H10 and H11 are distinct but related types of human nonpeptidergic neurons

Perhaps the most intriguing classes of human somatosensory neurons revealed by our transcriptomic approach are the H10 and H11 classes that primarily share features with the mouse nonpeptidergic nociceptors NP1–3 (*Figures 2 and 3*, *Figure 2—figure supplement 2*, *Figure 5—figure supplement 1*). ISH showed that the H10 and H11 classes of neurons, identified by their expression of *OSMR*

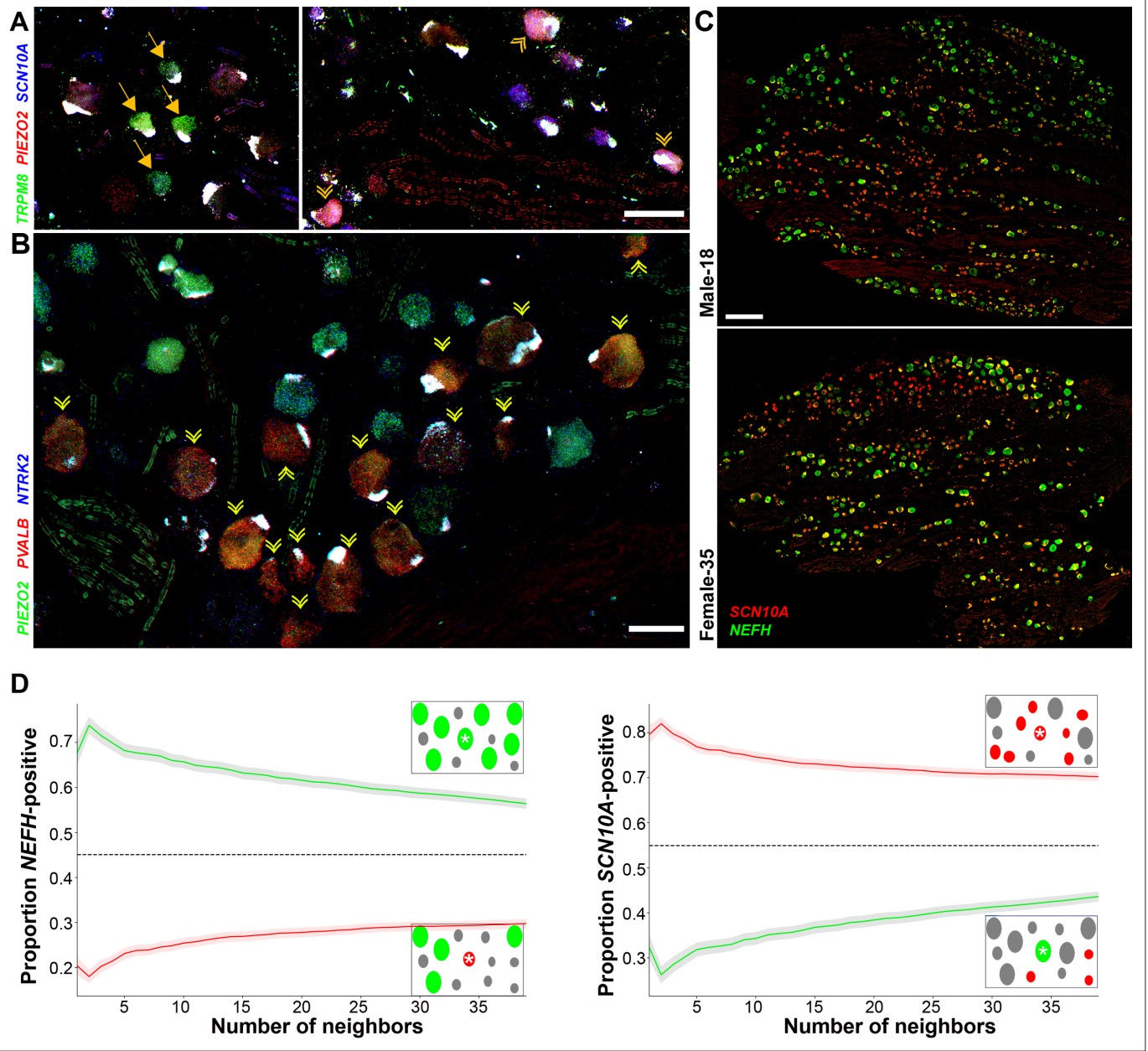

**Figure 4.** Transcriptomically related classes of human dorsal root ganglia (DRG) neurons are spatially clustered in the ganglion. Confocal images of sections through a human DRG probed for expression of key markers using multiplexed in situ hybridization (ISH); see **Figure 4—figure supplement 1** for the individual panels and additional probes. (**A**) Left panel shows a group of four H8-neurons (yellow arrows) that express *TRPM8* (green) but not *PIEZO2* (red) or *SCN10A* (blue). By contrast, right panel shows a different region of the ganglion where three H9-neurons coexpress these three transcripts (double arrowheads). (**B**) Other regions of the ganglia were dominated by larger diameter neurons. Putative proprioceptors, highlighted by double arrowheads, expressing *PIEZO2* (green) and *PVALB* (red), but not *NTRK2* (blue) were typically highly clustered in the ganglion. (**C**) Lower magnification images of complete sections stained for *NEFH* (green) and *SCN10A* (red) highlight the extensive co-clustering of large and small diameter neurons in different individuals. Scale bars = 100 μm in (**A** and **B**); 500 μm in (**C**). (**D**) The nearest *n* neighbors of *NEFH* and *SCN10A* single positive neurons in (**C**) were identified (for *n* = 1–40): green lines represent proportion (mean, solid line ± standard error of mean [SEM], shaded) of cells surrounding *NEFH*-positive neurons; red lines, proportion (mean, solid line ± SEM, shaded) of cells surrounding *SCN10A*-positive neurons. Left panel: proportion of surrounding cells that were *NEFH* positive. Right panel: proportion of surrounding cells that were *SCN10A* positive. Dashed black lines are the proportions expected for randomly distributed cells. Insets schematically show a central cell (highlighted by a star) and the surrounding neurons. Neighboring *NEFH*-positive cells are colored green and *SCN10A*-positive cells are colored red when these are being scored in the associated graph; gray cells are positive for the other marker. Clustering was statistically significant across the complete range (1–40 neighbors), p ≤ 6.96 × 10^{-42} (one-tailed Mann–Whitney *U*-test); *n* = 803 single positive cells confirming both short and long-range grouping of similar classes of human DRG neurons.

*Figure 4 continued on next page*

*Figure 4 continued*

The online version of this article includes the following figure supplement(s) for figure 4:

**Figure supplement 1.** Expression profiles of clusters of H8 (cool), H9 (human-specific), and H15 (proprioceptive) neurons.

**Figure supplement 2.** Expression profiles of clusters of H8 (cool), H9 (human-specific), and H15 (proprioceptive) neurons.

were small diameter neurons comparable in size to the *TAC1*-expressing peptidergic nociceptors (*Figure 5A*). Two qualitatively different ratios of *SCN10A* and *OSMR* were apparent in these cells (*Figure 5A*) hinting at their distinct identities. Our data (*Figure 2*, *Figure 2—figure supplement 2*, *Figure 5—figure supplement 1A*) show that H10 and H11 neurons express a number of genes that are known markers of mouse NP3 cells and functionally important for triggering pruritic responses (*Gatto et al., 2019*; *Solinski et al., 2019*). They are also distinguished from each other by expression of genes that likely play roles in itch and other aspects of somatosensation (*Figure 5B*, *Figure 2—figure supplement 2*, *Figure 5—figure supplement 2*). For example, although not prominently

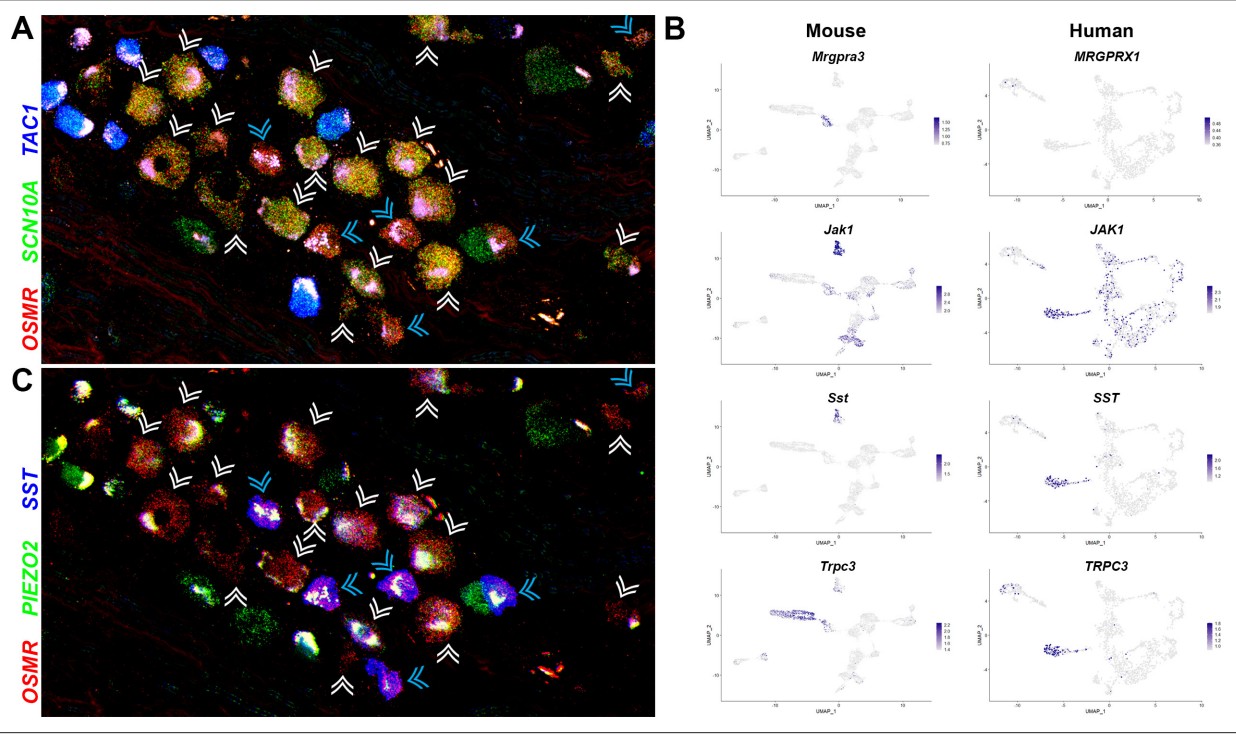

**Figure 5.** Two related classes of human nonpeptidergic small diameter neurons that may mediate itch. (**A**) Confocal image of a section through a human dorsal root ganglia (DRG) probed for nociception-related genes using multiplexed in situ hybridization (ISH). In this view, many neurons expressing the itch-related transcript, *OSMR* (red), are grouped together (arrowheads); cyan arrowheads point to cells that express relatively higher levels of *OSMR* than *SCN10A* (green). Peptidergic nociceptors marked by expression of *TAC1* (blue) and additional *SCN10A*-positive cells are also present in this region of the ganglion. (**B**) Universal manifold (UMAP) representation of mouse and human DRG neurons showing relative expression level (blue) of genes that distinguish H10 and H11 and mark-specific sets of mouse NP1–3 neurons. *MRGPRX1* is the human chloroquine receptor and the functional equivalent of *Mrgpra3*, which in mice marks NP2 cells. Note that coexpression patterns of *SST* and *JAK1* in H11 neurons resembles their expression in mouse NP3 pruriceptors but the ion channel *TRPC3* which also marks these cells is primarily expressed in mouse NP1 neurons; see *Figure 5—figure supplement 1* for additional breakdown of similarities and differences between nonpeptidergic neurons in mice and humans. (**C**) Confocal image of the group of candidate pruriceptors shown in (**A**) probed for expression of genes that distinguish H11 (*SST*, blue) from H10 cells (*PIEZO2*, green); note that neurons highlighted with cyan arrowheads have gene expression expected for H11 cells, whereas some H10 cells express lower levels of *SST* and also exhibit variation in the level of *PIEZO2* expression. Scale bars = 100 μm; see *Figure 5—figure supplement 1B* for individual channels and for expression of additional markers.

The online version of this article includes the following figure supplement(s) for figure 5:

**Figure supplement 1.** Expression profiles of human H10 and H11 dorsal root ganglia (DRG) neurons.

**Figure supplement 2.** Gene expression patterns in H10 and H11 classes of human dorsal root ganglia (DRG) neurons are distinct from classes of mouse small diameter nonpeptidergic neurons.

expressed, the human chloroquine responsive receptor *MRGPRX1* (*27*) localized selectively to H10 neurons (*Figure 5B*) perhaps suggesting a relationship to mouse NP2 cells. By contrast, Janus kinase 1 (*JAK1*), a mediator of itch through various types of cytokine signaling, including through OSMR (*Oetjen et al., 2017*), and the neuropeptide *SST* are particularly strongly expressed in H11 cells (*Figure 5B*). Both these genes are prominent markers of NP3 pruriceptors in mouse (*Figure 5B*). However, not all known itch-related transcripts are expressed in H10 and H11 neurons and both classes of cells express genes that better define NP1 neurons in mice as well as other cell types (*Figure 5B*, *Figure 2—figure supplement 2*, *Figure 5—figure supplement 2*).

H10 cells are also distinguished from H11 and mouse pruriceptors by their prominent expression of the stretch-gated ion channel *PIEZO2* (*Figure 1C*, *Figure 2—figure supplement 2*). The coexpression of itch-related transcripts and this low threshold mechanosensor hint that H10 neurons may be responsible for the familiar human sensation known as mechanical itch. However, their relationship to NP1-neurons revealed by co-clustering mouse and human data (*Figure 3*) and their expression of markers for various other cell types (*Figure 2—figure supplement 2*, *Figure 5—figure supplement 2*) possibly including nonpeptidergic cLTMRs suggest that their role in somatosensation may not be limited to itch alone.

A problem with single-cell sequencing approaches is the sparse nature of the data making it difficult to disentangle expression level from proportional representation in any cluster. This means that except for the most highly expressed genes, there is inherent ambiguity in interpreting the expression patterns. ISH provides an independent and more analog assessment of expression level that can help resolve this issue. Multiplexed ISH showed that *SST* divides the *OSMR*-positive cells into two intermingled types (*Figure 5C*) in keeping with the sequence data (*Figure 5B*) and the relative expression patterns of *SCN10A* and *OSMR* (*Figure 5A*). Moreover, the prediction that *PIEZO2-OSMR* coexpression should mark *SST*-negative neurons was also largely borne out by ISH (*Figure 5C*, *Figure 5—figure supplement 1*). However, ISH also shows that some neurons expressing lower levels of *SST* are *PIEZO2*-positive and that some *OSMR*-positive H10 cells, contain only a very low level of the mechanosensory channel (*Figure 5C*, *Figure 5—figure supplement 1*). Therefore, H10 and H11 are by no means homogeneous populations and may not be as functionally distinct as snRNA clustering suggests.

## Discussion

Transcriptomic analysis of DRG neurons confirms that mouse and human somatosensory neurons express many of the same genes (*Ray et al., 2018*). However, although gross similarity in the transcriptomic classification of these cells can be discerned in single-cell analyses (peptidergic versus nonpeptidergic; neurofilament rich, myelinated versus nonmyelinated), the patterns of coordinated gene expression across species are not well conserved and both species exhibit unique specializations. Recently, available transcriptomic data from the macaque further highlights the individuality of somatosensory neurons across species (*Kupari et al., 2021*). Surprisingly, in that study, despite major differences in gene expression between the monkey and mouse cell types, co-clustering approaches found an apparently close relationship between them including identification of large and distinct groups of NP1, NP2, and NP3 cells and putative cLTMRs (*Kupari et al., 2021*). However, perhaps because of fragility of large diameter neurons in cell-based droplet sequencing approaches, no candidate Aβ neurons and only very few Aδ cells were recovered (*Kupari et al., 2021*). By contrast our transcriptomic data and analysis combined with multiplexed ISH provide strong evidence for similarities between large diameter neurons between humans and mice but major differences in small diameter nonpeptidergic neurons as well as the existence of other human-specific cell types. Since in humans all nonpeptidergic neurons express *TRPV1* whereas mouse NP1, NP2, and cLTMRs do not prominently express this gene, this fits well with recent data showing much broader expression of *TRPV1* in humans than mice (*Shiers et al., 2020*).

At one level, this type of interspecies variation was unexpected given that there is similarity between the neuronal classes that comprise the mouse lumbar DRGs and trigeminal ganglia despite their very different types of innervation targets (*Sharma et al., 2020*; *Nguyen et al., 2019*). However, large changes in the receptive repertoire of other sensory systems have been observed and are thought to play a role in adaptation to specific ecological niches (*Yarmolinsky et al., 2009*). Thus, the evolution of DRG receptor cell diversity further highlights the importance of appropriate sensory input for fitness

and survival of a species. What is unusual relative to other senses is that transcriptomic differences are not limited to just the receptor repertoire for sensing environmental stimuli but instead extend to genes involved in the development and maintenance of defined neuronal subtypes. It is possible that this reflects major differences between mouse and human skin including fur covering. From a translational viewpoint, these differences could explain some of the problems in replicating results from mouse-based therapies (*Mogil, 2019*; *Yezierski and Hansson, 2018*) in humans and the availability of the human data may help direct research toward new targets and even suggest precision medicine strategies (e.g., to treat cold pain).

Our analysis identified particularly surprising differences between small diameter nonpeptidergic neurons in mice and humans (H10 and H11 cell types). In mice, one distinctive subset of these cells are the cLTMRs that innervate hairy skin and are thought to be responsible for affective touch (*McGlone et al., 2014*). At a transcriptomic level, humans do not have a clearly identifiable correlate for these cells (*Figure 3B*) although careful microneurography has revealed human c-fibers that respond to stroking (*Wessberg et al., 2003*). We suspect that some of these stroking responsive cells may be the nonpeptidergic H10 neurons that express itch-related genes as well as high levels of *PIEZO2*. In keeping with this suggestion, a recent preprint of a spatial transcriptomic analysis of human somatosensory gene expression (*Tavares-Ferreira et al., 2021*) designates cLTMRs as a subset of cells resembling H10 neurons that appear to have lower expression of some pruriceptive markers. However, it is also possible that human cLTMRs are other *PIEZO2*-expressing neurons that are unique to humans (*Figure 2B*, *Figure 1—figure supplement 3*). H10 and H11 neuron classes have more clear similarity to the mouse NP1–3 neurons but also exhibit major differences to all three types of cells. For example, in mice NP1 cells express a large combination of diagnostic markers (*Figure 5—figure supplement 2*) including *Mgrprd* that we did not find in our sequencing of human ganglion neurons. Bulk sequencing studies have identified *MRGPRD* expression in human DRG (*Price et al., 2016*), but recent ISH localization studies suggest only very low-level expression of this transcript together with *MRGPRX1* in somatosensory neurons (*Klein et al., 2021*). This would fit with our co-clustering that identifies the *MRGPRX1*-expressing H10 neurons as related to NP1 cells. However, many of the other NP1 markers have potential roles in signal detection and transduction but are not H10 selective (*Figure 5—figure supplement 2*). Moreover, in mice, *Mrgpra3* (the functional equivalent of *MRGPRX1*) marks the distinct NP2 neurons.

Taken together, our results and analysis suggest that experiments in mice are likely to illustrate general principles that are important for sensory detection and perception in humans but also imply that specific details related both to genes and cell-type responses may differ. In future studies, the central projections and targets of human somatosensory neuron subtypes might provide independent approaches for inferring function. Similarly, using immunohistochemistry (IHC) to understand how these cell classes innervate the skin and other tissues may allow correlation of arborization patterns with microneurography results. Since microneurography can be complemented by microstimulation this could ultimately reveal the role of specific neuronal classes in sensory perception (*Ackerley and Watkins, 2018*).

Our data provide a searchable database for gene expression in human DRG neurons. However, there are some limitations to the data and interpretation. For example, neither the number of neurons sequenced, nor the depth of sequencing is as comprehensive as for mice (*Sharma et al., 2020*; *Nguyen et al., 2019*). This means that rare neuronal subtypes and the expression patterns of even some moderately expressed genes may not be clear therefore it is likely that future studies and larger samples will be needed to refine these issues. Nonetheless, highly multiplexed ISH (*Figures 2 and 5*) confirm the major findings both about cell types and also gene expression and therefore substantiate the overall value of the data. The nuclear-based sequencing approach used here has advantages in preventing gene expression changes during single-cell isolation and is also likely to be less biased than cell-based approaches in terms of representation of the different cell types (*von Buchholtz et al., 2020*). However, sn-RNA sequencing provides a somewhat distorted view of cellular gene expression, as has been described for sensory neurons in mice (*Nguyen et al., 2019*). Therefore, it will be important to confirm expression levels of specific genes using complementary approaches. Finally, any functional roles for neuronal classes identified here have been extrapolated from expression of markers and distant similarity to mouse counterparts. Given the extensive differences that we report, some of these conclusions may need to be revised once cell class can be

linked to neuronal function in human subjects and/or using physiological tools in vitro with human tissues.

# Materials and methods

## Key resources table

| Reagent type (species) or resource | Designation | Source or reference | Identifiers | Additional information |
|---|---|---|---|---|
| Strain, strain background (*Mus musculus*) | C57BL/6NCrl | Charles River | Strain code: 027 | Male and female |
| Antibody | anti-NeuN (rabbit polyclonal) | Millipore | Cat#ABN78 | (1:4000 for nuclei isolation) (1:1000 for IHC) |
| Antibody | anti-Tubb3 (mouse monoclonal) | Proteintech | Cat#66375-1-Ig | (1:500) |
| Antibody | anti-rabbit Cy3-conjugated (donkey polyclonal) | JacksonImmuno Research | Cat#711-166-152 | (1:1000) |
| Antibody | anti-mouse FITC-conjugated (donkey polyclonal) | JacksonImmuno Research | cat#715-096-150 | (1:1000) |
| Sequence-based reagent | Chromium Next GEM Single Cell 3′ GEM, Library & Gel Bead Kit v3.1 | ×10 Genomics | Cat#1000128 | |
| Sequence-based reagent | Chromium Next GEM Chip G Single Cell Kit | ×10 Genomics | Cat#1000127 | |
| Commercial assay or kit | RNAscope HiPlex8 Detection reagents | Advanced Cell Diagnostics | Cat#324110 | |
| Commercial assay or kit | RNAscope HiPlex12 Ancillary reagents | Advanced Cell Diagnostics | Cat#324120 | |
| Commercial assay or kit | Human RNAscope probes (HiPlex 12) | Advanced Cell Diagnostics | *NEFH* (cat# 448141); *TRPM8* (cat# 543121); *PIEZO2* (cat# 449951); *SCN10A* (cat# 406291); *NTRK2* (cat# 402621); *TAC1* (cat# 310711); *OSMR* (cat# 537121); *SST* (cat# 310591); *TRPV1* (cat# 415381); *PVALB* (cat# 422181) | |
| Commercial assay or kit | RNAscope Fluorescent Multiplex assay | Advanced Cell Diagnostics | Cat #320851 | |
| Commercial assay or kit | Mouse RNAscope probes (MultiPlex) | Advanced Cell Diagnostics | *Scn10a* (cat#426011); *Nefh* (cat#443671) | |
| Commercial assay or kit | Human RNAscope probes (MultiPlex) | Advanced Cell Diagnostics | *OSMR* (cat#537121); *SST* (cat# 310591); *HRH1* (cat#416501); *NPPB* (cat#448511); | |
| Commercial assay or kit | NeuroTrace Green | Fisher Scientific | Cat#N21480 | (1:100) |
| Software, algorithm | CellRanger | ×10 Genomics | Version 2.1.1 | GRCh38.v25. premRNA |
| Software, algorithm | Seurat | Satija lab | Versions 3-4.04 | https://satijalab. org/seurat/ |
| Software, algorithm | Python | python.org | Version 3.7. | |
| Software, algorithm | scipy.stats | scipy.org | Version 1.5.2 | |
| Software, algorithm | scikit-learn | scikit-learn.org | Version 0.23.2 | |
| Software, algorithm | RStudio | https://www.rstudio.com/ | Version 1.4.1106 | |
| Software, algorithm | R | https://www.r-project.org/ | R version 4.1.1 | |
| Software, algorithm | DoubletFinder | https://github.com/chris-mcginnis-ucsf/DoubletFinder | | McGinnis lab (**McGinnis, 2021**) |
| Software, algorithm | ImageJ | http://imagej.nih.gov/ij | ImageJ 1.53 c | |
| Software, algorithm | Adobe Photoshop | https://www.adobe.com/ | 25.5.1 release | |
| Software, algorithm | kl_divergence | https://gist.github.com/lars-von-buchholtz/636f542ce8d93d5a14ae52a6c538ced5636f542ce8d93d5a14ae52a6c538ced5 | | |

*Continued on next page*

*Continued*

| Reagent type (species) or resource | Designation | Source or reference | Identifiers | Additional information |
|---|---|---|---|---|
| Other | RNA-later | Thermo Fisher | Cat# AM7021 | |
| Other | Spectrum Bessman tissue pulverizer | Fisher Scientific | Cat# 08-418-3 | |
| Other | Dounce homogenizer | Fisher Scientific | Cat# 357538 | |
| Other | 40 µm cell strainer | Thermo Fisher | Cat# 08-771-1 | |
| Other | Low bind microfuge tubes | Sorenson BioScience | Cat# 11,700 | |
| Other | SUPERaseIn RNase inhibitor | Thermo Fisher | Cat# AM2696 | 0.2 U/µl |
| Other | anti-rabbit IgG microbeads | Miltenyi biotec | Cat# 130-048-602 | |
| Other | LS column | Miltenyi biotec | Cat# 130-042-401 | |
| Other | Mouse DRG dataset | Woolf lab | GSE154659 | *Renthal et al., 2020*, Neuron |

## Study design

Transcriptomic analysis of human DRG neurons was carried out to establish similarities and differences between human somatosensory neurons and their counterparts in model organisms and to provide a resource. We chose a nuclear-based strategy because of its simplicity and quantitative nature relative to isolation of cells (*Nguyen et al., 2019*). All tissue was obtained from de-identified organ donors and was not preselected or otherwise restricted according to health conditions. We used DRGs from both male and female donors for the sequencing and ISH localization experiments. Randomization and blinding were not used because of the nature of our experiments. Similarly, before starting this study, we had no relevant information for setting sample size for snRNA sequencing from human DRG neurons. Therefore, we stopped data collection when we empirically determined that the cost of adding extra data outweighed the benefit of additional sequencing. In essence, numbers of sn-transcriptomes analyzed were limited by the availability of material and the difficulty of isolating human DRG nuclei with preservation of their transcriptome. We considered that the dataset would serve as a valuable and relatively comprehensive resource once including additional material from an individual preparation made only minor differences to the pattern of clustering we observed. Criteria for data exclusion followed standards in the field (see below) and sample sizes and numbers of replicates are also typical for this type of study and are described in the relevant experimental sections. Apart from the exclusions described for sn experiments, all data obtained were included in our study.

## Isolation of human DRG nuclei

DRG recovery was reviewed by the University of Cincinnati IRB #00003152; Study ID: 2015-5302, title Human dorsal root ganglia and was exempted. Lumbar L4 and L5 DRGs were recovered from donors withing 90 min of cross-clamp (*Valtcheva et al., 2016*). For ISH and IHC, DRGs were immersion fixed in 4% paraformaldehyde in phosphate-buffered saline (PBS) overnight, cryoprotected in 30% sucrose and were frozen in Optimal Cutting Temperature compound (Tissue Tek). For RNA sequencing, human DRGs immediately were cut into 1–2 mm pieces and stored in RNA-later (Thermo Fisher, Cat# AM7021). Excess RNA-later was removed and the tissue was frozen on dry ice and stored at −80°C. Nuclei were isolated from each donor separately as described previously (*Nguyen et al., 2019*) with minor modification. Briefly, the tissues were homogenized with a Spectrum Bessman tissue pulverizer (Fisher Scientific, CAT# 08-418-3) in liquid nitrogen. The sample was then transferred to a Dounce homogenizer (Fisher Scientific, Cat# 357538) in 1 ml of freshly prepared ice-cold homogenization buffer (250 mM sucrose, 25 mM KCl, 5 mM $MgCl_2$, 10 mM Tris, pH 8.0, 1 µM DTT, 0.1% Triton X-100 [vol/vol]). To lyse cells and preserve nuclei homogenization used 5 strokes with the 'loose' pestle (A) and 15 strokes with the 'tight' pestle (B). The homogenate was filtered through a 40 µm cell strainer (Thermo Fisher, cat# 08-771-1), was transferred to low bind microfuge tubes (Sorenson BioScience, cat# 11700) and centrifuged at 800 g for 8 min at 4°C. The supernatant was removed, the pellet gently resuspended in 1 ml of PBS with 1% bovine serum Albumin (BSA) and SUPERaseIn RNase Inhibitor (0.2 U/µl; Thermo Fisher, Cat#AM2696) and incubated on ice for 10 min.

Neuronal nuclei selection was performed by incubating the sample with a rabbit polyclonal anti-NeuN antibody (Millipore, cat#ABN78) at 1:4000 dilution with rotation at 4°C for 30 min. The sample was then washed with 1 ml of PBS with 1% BSA and SUPERaseIn RNase Inhibitor and centrifuged at 800 × $g$ for 8 min at 4°C. The resulting pellet was resuspended in 80 µl of PBS, 0.5% BSA, and 2 mM EDTA. 20 µl of anti-rabbit IgG microbeads (Miltenyi biotec, cat# 130-048-602) were added to the sample followed by a 20-min incubation at 4°C. Nuclei with attached microbeads were isolated using an LS column (Miltenyi Biotec, cat# 130-042-401) according to the manufacturer's instruction. The neuronal nuclei enriched eluate was centrifuged at 500 × $g$ for 10 min, 4°C. The supernatant was discarded, and the pellet was resuspended in 1.5 ml of PBS with 1% BSA. To disrupt any clumped nuclei, the sample was homogenized on ice with an Ultra-Turrax homogenizer (setting 1) for 30 s. An aliquot was then stained with trypan blue and the nuclei were counted using a hemocytometer. The nuclei were pelleted at 800 × $g$, 8 min at 4°C and resuspended in an appropriate volume for ×10 chromium capture. A second count was performed to confirm nuclei concentration and for visual inspection of nuclei quality. Note that prior to sequencing no check was made that this selection procedure was unbiased but we have previously used the approach to purify mouse trigeminal ganglion neuron nuclei (**Nguyen et al., 2019**) and have demonstrated highly quantitative recovery of all neuronal types (**von Buchholtz et al., 2020**). IHC analysis (**Figure 1—figure supplement 1**) supports the use of NeuN enrichment as a means to purify human DRG neurons and together with ISH quantitation (**Figure 2—figure supplement 1**) substantiates the relatively unbiased purification of various classes of neuronal nuclei using this method.

## Single nuclear capture, sequencing, and data analysis

10x-chromium capture and library generation were performed according to the manufacturer's instructions using v3 chemistry kits. Next generation sequencing was performed using Illumina sequencers. 10x chromium data were mapped using CellRanger to a pre-mRNA modified human genome (GRCh38.v25.premRNA). Data analysis used the Seurat V3 packages developed by the Satija lab and followed standard procedures for co-clustering (**Stuart et al., 2019**). For sn-RNA sequencing experiments cell filtering was performed as follows: outliers were identified and removed based on the number of expressed genes (500–10,000 retained) and mitochondrial proportion (<10% retained). Normalization and variance stabilization used regularized negative binomial regression (sctransform). After initial co-clustering of data from the different preparations, non-neuronal cell clusters

**Table 1.** Sequencing data for the five individual DRG nuclear preparations.

| Preparation# | DRG1-F36 | DRG2-M36 | DRG3a-F34 | DRG3b-F34 | DRG4-F35 | DRG5-F55 |
|---|---|---|---|---|---|---|
| Number of reads | 273,123,313 | 121,333,026 | 105,113,253 | 101,427,234 | 90,306,793 | 94,069,716 |
| Valid barcodes | 97.50% | 97.60% | 98.20% | 96.10% | 93.60% | 97.90% |
| Sequencing saturation | 93.20% | 93.10% | 39.70% | 82.10% | 38.70% | 55.90% |
| Q30 bases in barcode | 96.30% | 97.40% | 97.80% | 97.50% | 97.70% | 96.80% |
| Q30 bases in RNA read | 88.30% | 85.90% | 87.40% | 91.50% | 93.40% | 92.20% |
| Q30 bases in sample index | 96.00% | 96.80% | 97.40% | 96.60% | 96.60% | 94.80% |
| Q30 bases in UMI | 95.50% | 97.00% | 97.60% | 97.00% | 97.20% | 96.10% |
| Reads mapped confidently to genome | 86.50% | 81.20% | 71.20% | 16.90% | 10.80% | 18.10% |
| Reads mapped confidently to intergenic regions | 4.10% | 4.20% | 4.00% | 1.30% | 0.90% | 1.20% |
| Estimated number of cells | 584 | 273 | 6,180 | 223 | 872 | 999 |
| Mean reads per cell | 467,676 | 444,443 | 17,008 | 454,830 | 135,189 | 94,163 |
| Median genes per cell | 1917 | 709 | 786 | 481 | 872 | 812 |
| Total genes detected | 24,646 | 17,559 | 28,759 | 17,338 | 24,798 | 25,027 |
| Median UMI counts per cell | 2929 | 862 | 965 | 663 | 1,293 | 988 |
| DRG cells in final object | 212 | 152 | 770 | 80 | 281 | 342 |

were identified by their gene expression profiles. Clusters not expressing high levels of neuronal or somatosensory genes like *SNAP25*, *SCN9A*, *SCN10A*, *PIEZO2*, *NEFH*, etc. but instead expressing elevated levels of markers of non-neuronal cells including *PRP1*, *MBP*, *QKI*, *LPAR1*, and *APOE* were tagged as non-neuronal and were removed to allow reclustering of 'purified' human DRG neurons. A total of 1837 human DRG neuronal nuclei were included in the analysis (*Table 1*). The mean number of genes detected per nucleus was 2839 (range 501–9652), with a standard deviation of 1917. Doublet detection was performed on the individual datasets using DoubletFinder (*McGinnis et al., 2019*). For the clustering shown in the main figures the small number of potential doublets (*Figure 1—figure supplement 2E*) were not removed; principal components (PCs) were determined from integrated assay data and PCs 1–16 were used both for UMAP display of the data and for determining clusters. The resolution for clustering used relatively low stringency (2.0) and closely related clusters without distinguishing markers were merged. Changes in display clustering parameters and in the cutoffs for data inclusion/exclusion as well as leaving out nuclei from any single preparation made differences in how the data were represented graphically and the number of clusters identified but not to the main conclusions (*Figure 1—figure supplement 3*). All the different transcriptomically related neuron types described here could still be readily discerned in UMAP analysis of expression data. Identification of markers used the FindMarkers and FindAllMarkers functions of Seurat using default settings and with markers limited by a minimal level of expression in the positive cluster set using (min.pct = 0–0.3) and the difference in expression between positive and negative clusters set using (min.dif.pct = 0–0.3).

For analysis of the mouse, a random subset of data from sn-RNA sequencing of DRGs from wild type mice were extracted from data deposited by the Woolf lab (*Renthal et al., 2020*) using the R-function: sample. The data were filtered according to gene count (400–12,000 retained) and mitochondrial DNA (<1% retained) leaving 6895 DRG nuclei that were clustered using standard methods (*Stuart et al., 2019*). For the data that are displayed in most figures, PCs 1–20 were used for UMAP display with resolution for clustering set at 3.5; closely related clusters were merged. For *Figure 3—figure supplement 1*, PCs 1–30 were used for UMAP display with resolution for clustering set at 1.6; related clusters were not merged. The expression patterns that are described for genes in mice can also be checked in the outstanding and easy to search single-cell analysis provided by *Sharma et al., 2020*.

Co-clustering of mouse and human data used methods described by the Satija lab (*Stuart et al., 2019*). Briefly, we capitalized gene names in both mouse and human DRG-neuron data and then limited the datasets to genes that were shared between them. Mouse and human data were individually normalized and subjected to variance stabilization (sctransform) and were then combined using default parameters for integration (dims = 15). After integration, data were scaled and 30 PCs were calculated, used for UMAP display and clustering. Clustering resolution was set at 0.5 and cells were color coded by transferring their identity in the original clustering of human or mouse data. We experimented using different numbers of mouse neurons and found broadly similar results when approx. 1800, 3500, or 7000 mouse nuclei were used. However, although the relationships shown in *Figure 3* could be discerned using the lower numbers of mouse nuclei the mouse neurons were less organized (with some of the mouse clusters splitting). When we used substantially greater numbers of mouse neurons from the full Renthal dataset (*Renthal et al., 2020*), the mouse neurons dominated the clustering and only co-clustering of large diameter neurons across species was observed with the other human and mouse nuclei clustering separately from each other. In order to quantify the similarity/dissimilarity between a given mouse cluster and any human cluster, the KL divergence between their distributions in 30-dimensional continuous space (PCs 1–30 from the co-clustering) was estimated as described previously (*Perez-Cruz, 2008*) using R (https://gist.github.com/lars-von-buchholtz/636f542ce8d93d5a14ae52a6c538ced5; *von Buchholtz, 2021a*). The natural logarithm of the KL divergences for each mouse/human pairing was plotted as a heatmap in R.

## ISH and IHC

Cryosections from human DRGs (from different donors than those used for sequencing) were cut at 20 µm and used for ISH with the RNAscope HiPlex Assay (Advanced Cell Diagnostics) following the manufacturer's instructions. The following probes were used: *NEFH* (cat# 448141); *TRPM8* (cat# 543121); *PIEZO2* (cat# 449951); *SCN10A* (cat# 406291); *NTRK2* (cat# 402621); *TAC1* (cat# 310711); *OSMR* (cat# 537121); *SST* (cat# 310591); *TRPV1* (cat# 415381); and *PVALB* (cat# 422181). We also

used RNAscope Multiplex Fluorescent Assay (Advanced Cell Diagnostics) for localization of *OSMR* (cat# 537121); *NPPB* (cat# 448511); *HRH1* (cat# 416501); and *SST* (cat# 310591).

Confocal microscopy (5 μm spaced optical sections for HiPlex and 1 μm spaced optical sections for Multiplex Assays) was performed with a Nikon C2 Eclipse Ti (Nikon) using a ×40 objective. All confocal images shown are collapsed (maximum projection) stacks. HiPlex images were aligned and adjusted for brightness and contrast in ImageJ as previously described (*von Buchholtz et al., 2020*). Diagnostic probe combinations were used on at least three sections from at least two different individuals with qualitatively similar results. Overall, we used sections from ganglia from five different donors, however, the signal intensity of all probes varied between the individual ganglia making identification of positive signal risky in some cases. Strongest signals were observed for sections from an 18-year-old male donor and 23- and 35-year-old female donors. All images displayed here, and our analysis including cell counts were only from sections of ganglia isolated from these three individuals. When individual channels are displayed in the supplements, the strongest autofluorescence signals have been selected and superimposed using photoshop to help focus attention on ISH signal.

IHC was carried out using 20 μm sections of human DRGs using standard procedures. Primary antibodies to NeuN (Millipore) and β3 tubulin (Proteintech) were used at 1:1000 and 1:500 dilution, respectively, and were visualized using appropriate secondary antibodies conjugated to fluorescent reporters (Jackson Immuno Research, 1:1000). Sections were counterstained with 4',6-diamidino-2-phenylindole (DAPI) and/or green NeuroTrace (Fisher Scientific). Confocal images were acquired with a Nikon C2 Eclipse Ti (Nikon) using a ×40 objective with optical sections 5 μm apart. Images are maximum projection (collapsed) stacks of individual optical sections; consistency of NeuN staining was assessed using sections from two individuals; images were processed using Adobe Photoshop CC to adjust brightness, contrast, and set channel color for display.

Animal experiments were carried out in strict accordance with the US National Institutes of Health (NIH) guidelines for the care and use of laboratory animals and were approved by the National Institute of Dental and Craniofacial Research animal care and use committee (protocol #20-1041). Male and female mice were used for all experiments but were not analyzed separately. Mice were C57BL/6NCrl and were 6 weeks or older. Cryosections from fresh frozen mouse lumbar DRGs in OCT (Tissue-Tek) were cut at 10 μm and used for ISH with RNAscope Multiplex Fluorescent Assay (Advanced Cell Diagnostics) according to the manufacturer's instructions. Confocal images were acquired with a Nikon C2 Eclipse Ti (Nikon) using a ×10 objective at 1 μm optical section. Images are maximum projection (collapsed) stacks of 10 individual optical sections; consistency of staining was assessed using multiple sections from at least three mice as is considered standard; images were processed using Adobe Photoshop CC to adjust brightness, contrast, and set channel color for display.

## Spatial analysis of cell clusters

In order to quantify spatial clustering of cell types, neurons in ISH images (for humans, one male, one female and for mice several sections from different DRGs and animals) were manually outlined and annotated as *NEFH*-only or *SCN10A*-only; cells expressing both genes were not counted. Centroid coordinates of these cells and their distances were analyzed in Python 3.7. The nearest neighbors were identified based on Euclidean distance (Scikit-Learn package) and the percentage of *NEFH* and *SCN10A* cells in each neighborhood of size 1–40 cells was calculated. Statistical significance between *NEFH*- and *SCN10A*-surrounding neighborhoods was determined using a one-tailed Mann–Whitney *U*-test (Scipy Stats package).

## Acknowledgements

We thank the Genomics and Computational Biology Core (National Institute on Deafness and Other Communication Disorders) for sequencing; this work utilized the computational resources of the NIH HPC Biowulf cluster (http://hpc.nih.gov) and was supported in part by the Intramural Program of the National Institutes of Health, National Institute of Dental and Craniofacial Research. We are indebted to the LifeCenter, Cincinnati and the donor families for their generosity. We also thank Drs. Mark Hoon, Claire Le Pichon, and Alexander Chesler and members of our groups for valuable suggestions.

# Additional information

### Funding

| Funder | Grant reference number | Author |
|---|---|---|
| National Institute of Dental and Craniofacial Research | ZIC DE 000561 | Nicholas JP Ryba |
| National Institute of Neurological Disorders and Stroke | R01NS107356 | Steve Davidson |
| National Institutes of Health | RFNS113881 | Steve Davidson |

The funders had no role in study design, data collection and interpretation, or the decision to submit the work for publication.

### Author contributions

Minh Q Nguyen, Conceptualization, Data curation, Formal analysis, Investigation, Methodology, Writing – original draft, Writing – review and editing; Lars J von Buchholtz, Data curation, Formal analysis, Methodology, Software, Writing – original draft, Writing – review and editing; Ashlie N Reker, Investigation; Nicholas JP Ryba, Conceptualization, Data curation, Formal analysis, Funding acquisition, Investigation, Methodology, Supervision, Writing – original draft, Writing – review and editing; Steve Davidson, Conceptualization, Data curation, Funding acquisition, Investigation, Methodology, Writing – original draft, Writing – review and editing

### Author ORCIDs

Nicholas JP Ryba (iD) http://orcid.org/0000-0002-2060-8393
Steve Davidson (iD) http://orcid.org/0000-0003-2944-1144

### Ethics

Animal experiments were carried out in strict accordance with the US National Institutes of Health (NIH) guidelines for the care and use of laboratory animals and were approved by the National Institute of Dental and Craniofacial Research animal care and use committee (protocol #20-1041).

### Decision letter and Author response

Decision letter https://doi.org/10.7554/eLife.71752.sa1
Author response https://doi.org/10.7554/eLife.71752.sa2

# Additional files

### Supplementary files

• Transparent reporting form

### Data availability

Sequence data is available in GEO, accession number GSE168243; a searchable version of the data will also be made available at https://seqseek.ninds.nih.gov/home.

The following dataset was generated:

| Author(s) | Year | Dataset title | Dataset URL | Database and Identifier |
|---|---|---|---|---|
| Nguyen M, Ryba N, Davidson S | 2021 | Single nucleus transcriptomic analysis of human dorsal root ganglion neurons | https://www.ncbi.nlm.nih.gov/geo/query/acc.cgi?acc=GSE168243 | NCBI Gene Expression Omnibus, GSE168243 |

The following previously published datasets were used:

| Author(s) | Year | Dataset title | Dataset URL | Database and Identifier |
|---|---|---|---|---|
| Renthal W, Yang L, Tochitsky I, Woolf C | 2020 | Transcriptional reprogramming of distinct peripheral sensory neuron subtypes after axonal injury | https://www.ncbi.nlm.nih.gov/geo/query/acc.cgi?acc=GSE154659 | NCBI Gene Expression Omnibus, GSE154659 |

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
