## [Editor Report]

The manuscript by Nguyen et al. describes the assignment of neuronal cell types to human lumbar dorsal root ganglion (DRG) neurons based on the sequencing of individual nuclei. Bioinformatic comparison of these data to single nucleus sequencing results previously reported for mouse lumbar DRG is also described. The findings begin to close a gap in our understanding of how neuronal cell types and the expression of key genes differs between human and mouse. This kind of information is critical for targeted therapeutic efforts.

---

## [Decision Letter]

**Decision letter after peer review:**

Thank you for submitting your article "Single nucleus transcriptomic analysis of human dorsal root ganglion neurons" for consideration by *eLife*. Your article has been reviewed by 4 peer reviewers, one of whom is a member of our Board of Reviewing Editors, and the evaluation has been overseen by Catherine Dulac as the Senior Editor. The reviewers have opted to remain anonymous.

This is clearly an important and timely study of the gene expression profiles of individual human primary sensory neurons. The authors have acute access to quickly harvest tissue from donors which is extremely beneficial. While the number of neurons and depth of sequencing is on the low side, the authors provide new insights into the molecular organization as well as a comparison to mouse which will be valuable for the field. To be suitable for publication in *eLife*, the reviewers request that the manuscript is first strengthened by providing additional data, analysis and clarifications as outlined below.

Essential revisions:

1. There is concern that additional evidence is needed to support two major claims of the paper: that there are mouse-specific and human-specific neuron subtypes. These are issues stemming from the bioinformatics and from comparing datasets that were not generated in parallel. The in situ hybridization experiments provide some support for species specificity, but expression of individual marker genes may evolve while the overall cell population remains consistent. To address these issues, the authors should:

a) Add doublet detection;

b) Attempt different levels of clustering (number of clusters) and show the results are robust;

c) Try different methods and parameters of co-clustering across species;

d) Try a method of comparing across species that doesn't depend on the exact clusters same clusters found in the analysis. For example, this might involve training a classifier in mouse to predict cell type from gene expression. Then, applying it to human and showing that clearly conserved cell types show up, but that none of the species-specific populations are correctly labeled.

2. The reviewers had concern about the NeuN method of isolation. Was the technique in any way validated given the large number of non-neuronal cells and could there also have been bias in neurons that were selected by using this method, as it was pointed out that the levels of NeuN can vary across neuronal subtypes? Evidence validating the sorting and evidence against bias should be included, including plotting the distribution of UMIs and genes as well as using more than one glial marker.

3. The number of nuclei that were sequenced and the depth of sequencing are low. Can the authors make a short statement indicating that although this work enables a better understanding of the subsets of human sensory neurons, the readers should keep in mind that this classification will be further refined as more studies or larger samples are added

4. The authors should perform in situ for NPPB and also quantify all of the in situ data.

5. The execution of the experiment looking at organization of NEHF and SCN10A positive cells in the human DRG is not entirely compelling based on the few images shown but the quantification is potentially better. Depending on how strongly the authors want to claim the "spatial clustering" model, the authors should perform additional quantification/modeling and also similar analysis with mouse DRG neurons.

*Reviewer #3 (Recommendations for the authors):*

The most obvious factor to strengthen this paper would be to increase the number of nuclei used in the clustering analysis. This may be complicated by experimental issues, such as the need to run samples in parallel. They can perhaps make a short statement indicating that although this work enables a better understanding of the subsets of human sensory neurons, that the readers should keep in mind that this classification will be further refined as more studies or larger samples are added.

One other note is that there is less interest in the multiple comparisons to the mouse classification, but more on defining the sensory profiles in human. Just describing how one class of sensory neurons has the possibility to be polymodal based on its expression of different receptors, and build on that in the Results section. The part about adaptation to ecological niches in the discussion is exactly what this paper can bring that is unique and exciting.

For instance, I find this very interesting that sensory neurons innervating hairy vs. non-hairy skin neurons get prioritized for human somatosensation compared to the mouse.

On another note: do the authors know whether they aren't biasing their analysis for neurons of large or small diameter, they shouldn't be (since there isn't really a size filtration step) but maybe it would be worthwhile for them to do a sort of population study where they look at the overall diameters of neurons and generate ratios of large to medium to small diameter neurons in the human DRG and compare it to mouse. Then maybe comparing the markers used for a FISH to generate ratios of how much those cells occur in their snRNASeq data to confirm that they aren't accidentally biasing their analysis.

Figure 1: they need to state the number of nuclei analyzed in the text, not just the figure legend.

Figure 1: they should show a table with the number of nuclei from each donor.

Lines 379-381: they acknowledge that they have higher neuron recovery, but the tradeoff is lower read depth – wouldn't read depth be of the utmost priority when making claims about new classes of sensory neurons in humans?

Figure 2D: The authors claim that most neurons are labelled, yet from the image it seems like there are major gaps (that look like places where we'd find large diameter neurons based on the size). Did they use DAPI or any other counterstain at all to illustrate what the number of neurons actually was?

The authors claim they do not detect Mrgprd. Can they comment on the accuracy of snRNAseq at detecting low level transcripts? Is it that other low-level transcripts are undetectable? Can they comment on whether this could be a limitation of the technique?

Can the authors discuss the Shiers study published in Bioarchives from the lab of Ted Price, where they found that TRPV1 was broadly expressed?

Overall, I find this manuscript to be very important for the field of pain research and will be accessed frequently by pain scientists.

*Reviewer #4 (Recommendations for the authors):*

The number of cells and the depth of the sequencing are on the low end. One potential consequence is that populations such as c-LTRMs, which were reported by mouse studies and by the macaque and other human study are missing. Thus, it likely that there are not enough data to generate a comprehensive representation of all of the cell types. A bioinformatic comparison to the macaque would help to clarify some issues with what is different between these two important studies. In any case, a table that compares the cell types identified in the macaque and the other human study with this one would be very helpful. Information about the number of nuclei and "depth of sequencing" comparison across studies within this table or figure would also be helpful. The number of neuronal nuclei and genes per nuclei need to be stated upfront in the results.

The absence of C-LTMRs in the data is surprising. In mouse, C-LTMRs share features and genes with non-peptidergic neurons, are cooling sensitive and mechanically sensitive though the cooling sensitivity in mice is not likely mediated by Trpm8. How does the macaque c-LTMR cluster compare to the human data? How do the C-LTMRs assigned by the Price lab study compare?

For the comparison to mouse, the authors state that a random subset of Renthal data was used. Perhaps a better description of this would be helpful particularly what was in the subset? It is unclear why the full set would enrich for larger neurons.

Authors list genes that are not found in human data but are in mouse as well as a population that is in human that may not be in mouse. Page 8. The authors should use RT-PCR to see if the gene isn't expressed or if there is efficient nuclear export issue or annotation issue.

The authors should quantify the in situ data.

Confusing about how H5 has a mouse counterpart with a few different genes (single cell comparison) but then H9 doesn't have a mouse counterpart. Were computational methods used to compare to the single cell mouse data?

The logic of sentence on lines 217-220 escapes me. If the snRNA.Seq representation was not comprehensive I assume that you could get the same result – all cells have one of these genes and they are generally non-overlapping.

Organization to the DRG would make sense as it is a common theme for sensory systems and for the somatosensory system. The data are in the supplementary where the images are not entirely convincing (3-D reconstruction of 3 or 4 DRG would be more convincing) but the analysis seems to suggest this could be true. The authors suggest this organization is present in human but not mouse. The distribution of myelinated and unmyelinated should be shown for mouse using the same approach.

*Reviewer #5 (Recommendations for the authors):*

– It's a bit odd that quality of the NeuN sorting wasn't experimentally validated, especially given the large numbers of non-neuronal nuclei for a sorted experiment. The lack of structure in the non-neuronal cells could indicate that these droplets were empty or contained highly damaged cells. Plotting the distribution of UMIs and genes, as well as more than one glial marker could help resolve this.

– Levels of NeuN itself can vary across neurons subtypes (https://celltypes.brain-map.org/rnaseq/human_m1_10x). Could these be responsible for the differences in cell type across species?

– Seurat has different options for clustering included, each of them with a number of parameter options. There is not enough detail in the methods to identify if it was appropriate. In particular, how were the number of clusters identified? Do they really reflect distinct cell types or are there continuous signals? Similarly, the parameters outlier removal aren't described. These are especially critical for post-mortem experiments that can contain damaged cells.

– The length of time between subject death and sample collection should be included.

– There should be computation procedure for identifying/removing doublets.

– The number of and distribution of UMIs per nucleus in the human dataset should be reported.

– What parameters and function were used for the co-clustering between human and mouse?

– The KL divergence calculated identifies similarities for cell classes and seems to perform reasonably. However, I am not aware of that test being commonly used to link cell populations across species. What is the justification for using it.

– The procedure for identifying markers within and across species is not well-described. In particular, differences in the representation of different cell types across species could lead to artifacts in which markers are identified. Furthermore, since much of manuscript is devoted to differences between mouse and human expression, a statistical procedure should be used to verify the ability of particular genes to label a population in one species versus another.

[Editors' note: further revisions were suggested prior to acceptance, as described below.]

Thank you for resubmitting your work entitled "Single nucleus transcriptomic analysis of human dorsal root ganglion neurons" for further consideration by *eLife*. Your revised article has been reviewed by 3 peer reviewers and the evaluation has been overseen by Catherine Dulac as the Senior Editor, and a Reviewing Editor.

The manuscript has been significantly improved. There is one remaining issue that still needs to be further clarified or the claim tempered; namely there is concern that the authors have not convincingly proven that species specific cell types exist. Although the authors have argued the limitations of the bioinformatic approaches, it is suggested that they try generating substantially smaller clusters of cells in mouse (~twice the number of clusters). In this case, if none of these mouse sub-populations have a match to the human clusters based on KL divergence, that would substantially strengthen the argument that the human population is not conserved in mouse.

---

## [Author Response]

Essential revisions:1. There is concern that additional evidence is needed to support two major claims of the paper: that there are mouse-specific and human-specific neuron subtypes. These are issues stemming from the bioinformatics and from comparing datasets that were not generated in parallel. The in situ hybridization experiments provide some support for species specificity, but expression of individual marker genes may evolve while the overall cell population remains consistent. To address these issues, the authors should:a) Add doublet detection;

We added doublet detection Figure 1—figure supplement 2E.

b) Attempt different levels of clustering (number of clusters) and show the results are robust;

We added different clustering parameters to show robustness (Figure 1—figure supplement 3).

c) Try different methods and parameters of co-clustering across species;

Co-clustering of DRG data across species is a significant challenge because of the dramatic differences in gene expression even amongst the most conserved classes. As stated in the paper the Satija lab methodology was developed to find similarities across datasets without forcing matches. We also used CONOS: a program that provides significant flexibility for setting parameters. Using PCAs or PCCAs CONOS did not align mouse and human DRG data. With CCAs and parameters as used by Kupari et al., human and mouse DRG data appear to be closely related. However, these parameters inappropriately match human cortical excitatory and inhibitory neurons, DRG neurons with cortical excitatory neurons and even peripheral blood mononuclear cells with brain inhibitory neuron nuclei. Therefore, we have not included the analysis in our revised manuscript.

d) Try a method of comparing across species that doesn't depend on the exact clusters same clusters found in the analysis. For example, this might involve training a classifier in mouse to predict cell type from gene expression. Then, applying it to human and showing that clearly conserved cell types show up, but that none of the species-specific populations are correctly labeled.

It is not entirely clear what the reviewers are asking for; Kupari et al. trained a classifier with mouse data and then used it on macaque data (which seems to be what the reviewers want us to do), but this method forces a cell to “match” one of the mouse neuron types and has no control to ascertain validity. A serious concern is that all statistical evaluation of such an analysis assumes the basic tenet that testing data come from the same distribution as training data. This is clearly violated in across species comparisons. Moreover, the analysis does not evaluate closeness of relationships: the probability of a match is not a measure of similarity but of ambiguity between the limited prediction choices. For example, human cells in a cluster with gene expression features resembling three classes of neurons in mouse would distribute amongst the three clusters because of random differences at the single cell level. By contrast, cells in a cluster with very little similarity to any type of mouse neurons may be assigned to a single class because each cell needs to be matched to one of the mouse classes. We have not come up with an alternative that overcomes these problems. This is why we used Kullback-Leibler divergence to examine similarities between the robust clusters identified in mouse and human data. We now explain the appropriateness and rationale for using this method in the Results section (lines 234-241; line numbers refer to the word document with track changes all markup selected).

2. The reviewers had concern about the NeuN method of isolation. Was the technique in any way validated given the large number of non-neuronal cells and could there also have been bias in neurons that were selected by using this method, as it was pointed out that the levels of NeuN can vary across neuronal subtypes? Evidence validating the sorting and evidence against bias should be included, including plotting the distribution of UMIs and genes as well as using more than one glial marker.

We have used this approach in mice but had not tested if the NeuN method was biased in humans before use. However, we needed to deplete the roughly 100-fold excess of non-neuronal nuclei in human DRGs. We have now carried out immunohistochemistry using the NeuN antibody and demonstrate that the vast majority of neurons (but not other cells) are robustly stained in sections (new Figure 1—figure supplement 1). We believe this will be valuable information for readers and should assuage reviewers’ concerns.

3. The number of nuclei that were sequenced and the depth of sequencing are low. Can the authors make a short statement indicating that although this work enables a better understanding of the subsets of human sensory neurons, the readers should keep in mind that this classification will be further refined as more studies or larger samples are added

Statement added to discussion (lines 427-428)

4. The authors should perform in situ for NPPB and also quantify all of the in situ data.

*NPPB* expression is not a focus of this paper and is only mentioned once in passing in the results. Nonetheless, we obtained relevant probes and carried out this request. As expected from the sequence data and in agreement with previous ISH data published by other groups, our data demonstrate low-level expression of *NPPB* in H10 and H11 neurons (new Figure 5—figure supplement 1A).

At the reviewers’ request, we also include extensive quantitation of ISH data (new Figure 2—figure supplement 1). Despite recent trends to “quantitate” all ISH data, we are not keen on this level of analysis (unless it can be done by automated signal detection) because many factors can influence the scoring of positives. Extensive autofluorescence makes automated signal detection impractical for human DRGs, thus we concentrated on diagnostic genes.

5. The execution of the experiment looking at organization of NEHF and SCN10A positive cells in the human DRG is not entirely compelling based on the few images shown but the quantification is potentially better. Depending on how strongly the authors want to claim the "spatial clustering" model, the authors should perform additional quantification/modeling and also similar analysis with mouse DRG neurons.

This is not a major point of the paper but helps explain the anatomically clustered nature of rare but related cell-types including proprioceptors and “cool sensing” cells. Nearest neighbor analysis provides a strong argument for organization at the level of the large diameter non-nociceptive neurons (NEFH-only) and the smaller nociceptors (SCN10A-only): for up to 40 nearest neighbors of 803 single positive cells in 2 sections, the maximum p-value was < 7 x 10^-42^. We have moved this analysis to the main figure to highlight the quantitative measure. We carried out the same analysis in mice Figure 4—figure supplement 2 as requested by the referees. Our data demonstrate that a similar organization is also present there strengthening the conclusions by showing conservation across species.

Reviewer #3 (Recommendations for the authors):[…] Figure 1: they need to state the number of nuclei analyzed in the text, not just the figure legend.Figure 1: they should show a table with the number of nuclei from each donor.

We moved the number to the text and list the nuclei from each donor in the table.

Figure 2D: The authors claim that most neurons are labelled, yet from the image it seems like there are major gaps (that look like places where we'd find large diameter neurons based on the size). Did they use DAPI or any other counterstain at all to illustrate what the number of neurons actually was?

Neurons are well spaced in human DRGs and surrounded by nerve fibers and other structures; we provide an image in the supplement where the brightness/contrast has been adjusted to highlight this point. See also new Figure 1—figure supplement 1 for DAPI and other counterstaining.

The authors claim they do not detect Mrgprd. Can they comment on the accuracy of snRNAseq at detecting low level transcripts? Is it that other low-level transcripts are undetectable? Can they comment on whether this could be a limitation of the technique?

We already addressed the limitations of snRNAseq in the paper and explained that the low expression of *MRGPRD* in human DRG neurons was consistent with recently published ISH images for this transcript showing expression close to the detection limit of the technique. By contrast, in mouse *Mrgprd*-expression is very robust.

Can the authors discuss the Shiers study published in Bioarchives from the lab of Ted Price, where they found that TRPV1 was broadly expressed?

Our data and theirs are consistent; we have added a sentence to the discussion and reference to the now published Shiers paper (ref-43, lines 346-349).

Reviewer #4 (Recommendations for the authors):The number of cells and the depth of the sequencing are on the low end. One potential consequence is that populations such as c-LTRMs, which were reported by mouse studies and by the macaque and other human study are missing. Thus, it likely that there are not enough data to generate a comprehensive representation of all of the cell types. A bioinformatic comparison to the macaque would help to clarify some issues with what is different between these two important studies. In any case, a table that compares the cell types identified in the macaque and the other human study with this one would be very helpful. Information about the number of nuclei and "depth of sequencing" comparison across studies within this table or figure would also be helpful. The number of neuronal nuclei and genes per nuclei need to be stated upfront in the results.The absence of C-LTMRs in the data is surprising. In mouse, C-LTMRs share features and genes with non-peptidergic neurons, are cooling sensitive and mechanically sensitive though the cooling sensitivity in mice is not likely mediated by Trpm8. How does the macaque c-LTMR cluster compare to the human data? How do the C-LTMRs assigned by the Price lab study compare?

Note we do not think that humans have no cLTMRs merely that they don’t have an identifiable group of cells that correspond with the mouse neurons that are thought to play this role (see lines 391-400). We have added some discussion about the macaque study (lines 160-161 and 335-402) but the differences in approaches (e.g., cells vs nuclei) make extensive comparisons difficult. The Price lab study has not yet been published and the original bioRxiv version did not assign any neurons as cLTMRs. This was changed at a later time making it dangerous to extensively evaluate similarities and differences between our study and theirs since their interpretation may yet change. Therefore, we have removed most of the discussion we had about the spatial transcriptomic analysis but now point out (lines 397-400) that our speculation that H10 neurons may function as cLTMRs is consistent with their assignment of transcriptomically similar cells as cLTMRs.

For the comparison to mouse, the authors state that a random subset of Renthal data was used. Perhaps a better description of this would be helpful particularly what was in the subset? It is unclear why the full set would enrich for larger neurons.

We added more detail to the methods. The full Renthal dataset does not enrich for larger neurons but clusters in the same way as smaller subsets; we used the smaller group to prevent the mouse data overwhelming the human data in co-clustering as is now explained more carefully in the methods. The smaller subsets of mouse data (including just 1800 nuclei like our human dataset) are also (1) just as informative as the full dataset in terms of clustering and (2) more efficient to use computationally.

The logic of sentence on lines 217-220 escapes me. If the snRNA.Seq representation was not comprehensive I assume that you could get the same result – all cells have one of these genes and they are generally non-overlapping.

Thanks, we have rewritten the sentence to better explain our interest in these 3 genes.

Reviewer #5 (Recommendations for the authors):[…] – The number of and distribution of UMIs per nucleus in the human dataset should be reported.

Added to Figure 1—figure supplement 2D.

– What parameters and function were used for the co-clustering between human and mouse?– The procedure for identifying markers within and across species is not well-described. In particular, differences in the representation of different cell types across species could lead to artifacts in which markers are identified.

We have added more detail to the methods.

[Editors' note: further revisions were suggested prior to acceptance, as described below.]

The manuscript has been significantly improved. There is one remaining issue that still needs to be further clarified or the claim tempered; namely there is concern that the authors have not convincingly proven that species specific cell types exist. Although the authors have argued the limitations of the bioinformatic approaches, it is suggested that they try generating substantially smaller clusters of cells in mouse (~twice the number of clusters). In this case, if none of these mouse sub-populations have a match to the human clusters based on KL divergence, that would substantially strengthen the argument that the human population is not conserved in mouse.

We carried out the requested analysis (19 clusters rather than 10) and demonstrate that with increased granularity in the mouse data, the same relationships between human and mouse cell types are seen (new Figure 3—figure supplement 1). Importantly, neither subcluster of mouse cLTMRs is a good match for any type of human DRG-neurons. Similarly, the human neuronal classes that only poorly matched the mouse clusters were not better matched to any of the subdivided mouse cell types. Nonetheless we also toned down the abstract.